

# Psychometric properties of the Automatic Thoughts Questionnaire-8 in two Spanish nonclinical samples

Francisco J. Ruiz[1], Miguel A. Segura-Vargas[1], Paula Odriozola-González[2] and Juan C. Suárez-Falcón[3]

[1] Fundación Universitaria Konrad Lorenz, Bogotá, Colombia
[2] Universidad de Valladolid, Valladolid, Spain
[3] Universidad Nacional de Educación a Distancia (UNED), Madrid, Spain

## ABSTRACT

**Background**. The ATQ is a widely used instrument consisting of 30 items that assess the frequency of negative automatic thoughts. However, the extensive length of the ATQ could compromise its measurement efficiency in survey research. Consequently, an 8-item shortened version of the ATQ has been developed. This study aims to analyze the validity of the ATQ-8 in two Spanish samples.

**Method**. The ATQ-8 was administered to a total sample of 1,148 participants (302 undergraduates and 846 general online population). To analyze convergent construct validity, the questionnaire package also included the Dysfunctional Attitude Scale-Revised (DAS-R), Depression Anxiety and Stress Scale-21 (DASS-21), Acceptance Action Questionnaire-II (AAQ-II), Cognitive Fusion Questionnaire (CFQ), Generalized Pliance Questionnaire (GPQ), and Satisfaction with Life Scale (SWLS). To analyze internal consistency, we computed Cronbach's alpha and McDonald's omega. A confirmatory factor analysis was conducted to test the one-factor structure of the ATQ-8. In so doing, a robust diagonally weighted least square estimation method (Robust DWLS) was adopted using polychoric correlations. Afterward, we analyzed measurement invariance across samples, gender, groupage, and education level. Lastly, we evaluated convergent construct validity by computing Pearson correlations between the ATQ-8 and the remaining instruments.

**Results**. The internal consistency across samples was adequate (alpha and omega = .89). The one-factor model demonstrated a good fit to the data (RMSEA = 0.10, 90% CI [0.089, 0.112], CFI = 0.98, NNFI = 0.97, and SRMR = 0.048). The ATQ-8 showed scalar metric invariance across samples, gender, groupage, and education level. The ATQ-8 scores were significantly associated with emotional symptoms (DASS-21), satisfaction with life (SWLS), dysfunctional schemas (DAS-R), cognitive fusion (CFQ), experiential avoidance (AAQ-II), and generalized pliance (GPQ). In conclusion, the Spanish version of the ATQ-8 demonstrated adequate psychometric properties in Spanish samples.

Corresponding author
Francisco J. Ruiz,
franciscoj.ruizji@gmail.com

## INTRODUCTION

Unipolar depression is characterized by sadness, irritability, or anhedonia, as well as a loss of appetite, difficulty to sleep, fatigue, slowing of speech and action, and suicidal thoughts, among others (*American Psychiatric Association, 2013*). The cognitive model proposed by *Beck et al. (1979)* states that the cognitive triad, integrated by a pattern of negative thinking about the world, the future, and the self, is one of the pillars of depression. Within this cognitive pattern, negative automatic thoughts play a crucial role and are defined as negative self-statements (*Beck et al., 1979*).

The Automatic Thoughts Questionnaire is one of the most extensively used instruments to measure negative automatic thoughts (ATQ; *Hollon & Kendall, 1980*). The ATQ is an instrument consisting of 30 items with a 5-point Likert scale that assesses the frequency of negative automatic thoughts experienced during the past week. *Hollon & Kendall (1980)* asked 312 undergraduates to recall dysphoric experiences and to report associated cognitions. Afterward, the authors chose 100 representative cognitions and administered them to a second sample. Through a cross-validation analysis, the authors retained 30 of the 100 original items. These items significantly discriminated between clinical and nonclinical samples (*Hollon & Kendall, 1980*).

Several studies have confirmed the temporal consistency, convergent and discriminant validity, and excellent internal consistency of the ATQ (e.g., *Chioqueta & Stiles, 2004*; *Hollon & Kendall, 1980*; *Hollon, Kendall & Lumry, 1986*; *Kazdin, 1990*). The results of exploratory factor analyses across different studies yielded factor structures with more than one factor (e.g., *Bryant & Baxter, 1997*; *Cano-García & Rodríguez-Franco, 2002*; *Chioqueta & Stiles, 2006*; *Deardorff, Hopkins & Finch Jr, 1984*; *Ghassemzadeh et al., 2005*; *Joseph, 1994*; *Kazdin, 1990*; *Oei & Mukhtar, 2008*; *Sahin & Sahin, 1992*; see reviews in *Netemeyer et al., 2002*; *Zettle et al., 2013*). Most studies have obtained different factor solutions from the four factors shown by *Hollon & Kendall (1980)*. *Netemeyer et al. (2002)* mentioned that all studies found that a large proportion of the variance was accounted for the first factor. Consequently, the results suggest that one factor could underlie the 30 items of the ATQ. Moreover, most studies have only used the overall score of the ATQ, which treats the scale as if it were only represented by one factor.

The extensive length of the ATQ could compromise its measurement efficiency in survey research. Accordingly, *Netemeyer et al. (2002)* gathered two samples ($N = 434$ and $N = 419$) to derive the 15- and 8-item reduced versions of the ATQ. Both versions of the questionnaire had a single factor, with alphas of .96 and .92, respectively. Two additional cross-validation samples ($N = 163$ and $N = 91$) also showed support for the 15-and 8-item reduced versions, which suggests that the shortened versions of the ATQ are suitable alternatives to measure automatic cognitions associated with depression (*Netemeyer et al., 2002*).

Following the study by *Netemeyer et al. (2002)*, *Ruiz, Suárez-Falcón & Riaño Hernández (2017)* analyzed the psychometric properties of the Spanish version of the ATQ-8 in a Colombian sample of 1,587 participants, including general population, a clinical sample, and undergraduates. The analysis displayed good internal consistency across samples (alpha

of. 89), and the one-factor model obtained an adequate fit to the data (RMSEA = 0.083, 90% CI [0.074, 0.092]; CFI = .96; NNFI = .95). Additional factor analyses confirmed measurement invariance across gender and samples (i.e., clinical and nonclinical samples). Furthermore, the mean scores of the clinical sample were significantly higher than the scores of their nonclinical counterpart.

The results presented in *Netemeyer et al. (2002)* and *Ruiz et al. (2017b)* indicate that the ATQ-8 might be an excellent alternative to the original ATQ scale. However, the factor structure and psychometric properties of the ATQ-8 have been analyzed only in two countries. Accordingly, the current study aims to analyze the validity of the ATQ-8 in Spaniard samples. This study is relevant because the original ATQ was only preliminarily validated in Spain by *Cano-García & Rodríguez-Franco (2002)* in a sample of 205 individuals suffering from chronic pain. Thus, there is scarce empirical evidence of the psychometric properties of the ATQ in nonclinical samples in Spain.

This study analyzes the factor structure and psychometric properties of the ATQ-8 in two nonclinical Spanish samples. The first sample consisted of 302 undergraduates and the second one of 846 individuals from the general population.

## MATERIALS & METHODS

The procedures followed in the research reported in the manuscript were approved by the Bioethics Committee of Fundación Universitaria Konrad Lorenz (2016-021B). Written informed consent was obtained from all participants in this study.

### Participants

*Sample 1*. This sample consisted of 302 undergraduates (age range 18–61, $M = 26.18$, $SD = 9.75$, 64.6% of females) from a Spanish university. Of the overall sample, 4.3% of the participants were currently in treatment, 19.4% had received psychological or psychiatric treatment, and 3.7% were taking psychotropic medication.

*Sample 2*. This sample consisted of 846 participants from general population, who completed the instruments online (age range 18-72, $M = 35.14$, $SD = 11.39$, 75.7% of females). Of the overall sample, 3.4% of participants had completed primary studies, 31% secondary studies, and 55.6% were university graduates. Also, 12.8% of participants were currently in treatment, 44.6% had received psychological or psychiatric treatment, and 12.9% were taking psychotropic medication.

### Instruments

*Automatic Thoughts Questionnaire-8* (ATQ-8; *Netemeyer et al., 2002*; Spanish version by *Cano-García & Rodríguez-Franco, 2002*). The ATQ-8 is the reduced version of the ATQ. Through a Likert-type scale (5 = *all the time*; 1 = *not at all*), it measures the frequency of negative thoughts during the past week. Examples of items are "I'm so disappointed in myself", "I feel so helpless", "My future is bleak", and "I can't finish anything".

*Dysfunctional Attitude Scale-Revised* (DAS-R; *De Graaf, Roelofs & Huibers, 2009*; Spanish version by *Ruiz et al., 2015*). The DAS is a traditional instrument that measures dysfunctional schemas. Its revised version (i.e., DAS-R) has 17 items, which are responded

on a 7-point Likert-type scale (7 = *fully agree*; 1 = *fully disagree*), organized into two factors: Perfectionism/Performance evaluation and Dependency. Examples of the items are: "If a person asks for help, it is a sign of weakness", "My happiness depends more on other people than it does on me", "If I fail at my work, then I am a failure as a person", and "If others dislike you, you cannot be happy". The DAS-R has shown a factor structure with two correlated factors and a second-order factor and has also demonstrated adequate psychometric properties in Spanish and Colombian samples (*Ruiz et al., 2016*; *Ruiz et al., 2015*). In this study, the DAS-R presented a Cronbach's alpha of .88 in Sample 1. According to the cognitive model of depression, medium to strong correlations were expected between the DAS-R and the ATQ-8.

*Depression, Anxiety, and Stress Scales-21* (DASS-21; *Lovibond & Lovibond, 1995*; Spanish version by *Daza et al., 2002*). The DASS-21 measures negative emotional states experienced during the last week through 21 items on a 4-point Likert-type scale (3 = *applied to me very much or most of the time*; 0 = *did not apply to me at all*). Examples of the items are: "I couldn't experience positive feeling", "I felt close to panic", and "I found it difficult to relax". The DASS-21 has shown a hierarchical factor structure consisting of three first-order factors (Depression, Anxiety, and Stress) and a second-order factor. The latter can be considered as an overall indicator of emotional symptoms (*Ruiz et al., 2017a*). The DASS-21 has also presented good convergent and discriminant validity and internal consistency. Alpha values in this study for the DASS-Total were .92 and .95 for Sample 1 and 2, respectively. The DASS-21 was administered because, in previous studies, emotional symptoms and not only depression have been strongly associated with the frequency of negative thoughts. Consequently, strong correlations were expected between the DASS-21 subscales and the ATQ-8.

*Satisfaction with Life Scale* (SWLS; *Diener et al., 1985*; Spanish version by *Atienza et al., 2000*). The SWLS measures self-perceived well-being through 5 items, graded with a 7-point Likert-type scale (7 = *strongly agree*; 1 = *strongly disagree*). Examples of items are "If I could live my life over, I would change almost nothing", "In most ways, my life is close to my ideal", and "The conditions in my life are excellent". The SWLS has demonstrated adequate convergent validity and psychometric properties. Alpha values in the study were .84 and .89 for Samples 1 and 2, respectively. Previous research has demonstrated that the frequency of negative thoughts is negatively associated with life satisfaction (*Ruiz et al., 2017b*). Medium to strong negative correlations were expected between the SWLS and the ATQ-8.

*Acceptance and Action Questionnaire-II* (AAQ-II; *Bond et al., 2011*; Spanish version by *Ruiz et al., 2013*). The AAQ-II measures general experiential avoidance through 7 items and a 7-point Likert-type scale (7 = *always*; 1 = *never true*). The items evaluate the reluctance to experience unwanted emotions and thoughts as well as the inability to be in the present moment and behave towards value-directed actions when experiencing psychological discomfort. Examples of items are: "Emotions cause problems in my life", "I worry about not being able to control my worries and feelings", and "It seems like most people are handling their lives better than I am". The Spanish version by *Ruiz et al. (2013)* demonstrated a one-factor structure and good psychometric properties in Spanish

samples with an overall alpha of .88. Alpha values in this study were .91 for both Sample 1 and Sample 2. The AAQ-II was administered because prior research has obtained strong positive correlations between ATQ scores and the AAQ-II (e.g., *Ruiz & Odriozola-González, 2016*).

*Cognitive Fusion Questionnaire* (CFQ; *Gillanders et al., 2014*; Spanish version by *Ruiz et al., 2017b*). The CFQ measures cognitive fusion as averaged across contexts through 7 items and a 7-point Likert-type scale (7 = *always*; 1 = *never true*), where higher scores indicate a higher degree of cognitive fusion. Examples of the items are: "I over-analyze situations to the point where it's unhelpful to me", "I get upset with myself for having certain thoughts", and "I struggle with my thoughts". The English validation of the CFQ has demonstrated to have good reliability, temporal stability, sensitivity to treatment effects, a one-factor structure, and convergent, divergent, and discriminant validity. The Spanish translation has proven to have similar psychometric properties (alpha = .92) and factor structure to the original version (*Ruiz et al., 2017b*). In this study, the CFQ obtained alphas of .90 and .93 for Samples 1 and 2, respectively. Medium to strong positive correlations between the CFQ and the ATQ-8 were expected.

*Generalized Pliance Questionnaire* (GPQ; *Ruiz et al., 2019*). The GPQ is a questionnaire consisting of 18 items, graded on a 7-point Likert-type scale (7 = *always true*; 1 = *never true*) that measures generalized pliance, defined as a pattern of rule-governed behavior in which the individual's primary source of reinforcement is social whim. Examples of the items are: "I care a lot about what my friends think of me", "My main goal in life is to be recognized and respected by those around me", and "My decisions are very much influenced by other people's opinions". In this study, the GPQ obtained an alpha of .92 and .95 in Samples 1 and 2, respectively. Medium to strong positive correlations were expected between the GPQ and the ATQ-8.

## Procedure

For Sample 1, the instruments package was administered in the classrooms during a regular class. In Sample 2, participants answered an online survey that was advertised through social media (e.g., Facebook, institutional webpages, etc.). In both samples, participants provided written informed consent. Participants in Sample 1 responded to the following instruments: ATQ-8, DAS-R, DASS-21, SWLS, AAQ-II, CFQ, and GPQ. Participants in Sample 2 responded to the same questionnaires except for the DAS-R. Once the participants completed the study, the aims of the study were debriefed, and they were also thanked for their participation. No incentives were provided to the participants.

## Statistical and psychometric analysis

Before conducting factor analyses, the data from both samples were examined to find missing values. However, no missing data were found. Corrected item-total correlations were computed on SPSS 25© to find items that should be removed due to a low discrimination item index (i.e., values below .30). McDonald's omega and Cronbach's alpha were conducted to explore the ATQ-8 internal consistency with total sample ($N = 1148$) and providing percentile bootstrap confidence intervals (CI) (*Viladrich, Angulo-Brunet &*

 

*Doval, 2017*). The MBESS package in R was used to compute these coefficients (*Kelley & Lai, 2012*; *Kelley & Pornprasertmanit, 2016*).

Because the ATQ-8 is responded on a 5-point Likert-type scale, an estimation method appropriate for ordinal data was selected to conduct the CFA. Accordingly, a robust diagonally weighted least square estimation method (Robust DWLS) was adopted using polychoric correlations. These analyses were conducted with LISREL © (version 8.71, *Jöreskog & Sörbom, 1999*). For the one-factor model, the chi-square test and the following goodness of fit indexes were calculated: (a) the root mean square error of approximation (RMSEA), (b) the comparative fit index (CFI), (c) the non-normed fit index (NNFI), and (d) the standardized root mean squared residual (SRMR). SRMR values below 0.05 reflect a very good fit to the data and values of 0.08 reflect a good fit to the data (*Hu & Bentler, 1999*; *Kelloway, 1998*). *Kelloway (1998)* suggested that values of RMSEA of 0.10 represent an acceptable or modest fit, whereas Hu and Bentler reduced the value to 0.08. Nevertheless, both guidelines suggest that a value of 0.05 reflects a very good fit to the data. Regarding the CFI and NNFI, values above .95 show a good fit to the data and above .90 indicate adequate-fitting models.

Following *Jöreskog (2005)* and *Millsap & Yun-Tein (2004)*, additional CFAs were conducted to assess for metric and scalar invariances across samples, gender, groupage (younger or equal to 35 years vs. older than 35 years), and education level (primary and secondary studies vs. university studies). Metric invariance means that item factor loadings are invariant across samples, gender, groupage, and education level, whereas scalar invariance involves that item intercepts are also invariant. Consequently, a comparison was conducted among the relative fits of three increasingly restrictive models: the scalar invariance model, the metric invariance model, and the multiple-group baseline model. In so doing, we compared the relative fit of three increasingly restrictive nested models: the multiple-group baseline model (it allowed the unstandardized factor loadings to vary across groups), the metric invariance model (it placed equality of factor loadings across groups), and the scalar invariance model (it placed equality in both the factor loadings and the item intercepts across groups). For the comparison model, the indices of the CFI, NNFI, and RMSEA were compared among the nested models. Regarding the selection of a model, the more constrained model was carefully chosen (i.e., second model versus the first model, and third model versus the second model) if the following criteria proposed by *Cheung & Rensvold (2002)* and *Chen (2007)* were fulfilled: (a) the difference in RMSEA (∆RMSEA) was lower than .01; (b) the differences in CFI (∆CFI) and NNFI (∆NNFI) were higher or equal to -.01.

Descriptive data were also calculated. To explore gender differences in ATQ-8 scores, an independent $t$-test was computed. Lastly, to evaluate convergent construct validity, Pearson correlations between the ATQ-8 and the other instruments were calculated.

## RESULTS

### Descriptive data and psychometric quality of the items

Table 1 displays the Spanish translation of the items of ATQ-8 with their corrected item-total correlations for each sample and descriptive data. The eight items presented

**Table 1  Item description and corrected item-total correlations.**

| Item number and description | Corrected item-total correlation | | |
|---|---|---|---|
| | Sample 1 Undergraduates | Sample 2 General population online | Overall sample |
| 1. No soy Bueno [I'm no good]. | .51 | .58 | .56 |
| 2. "Soy tan decepcionante hasta para mí mismo" [I'm so disappointed in myself]. | .72 | .74 | .74 |
| 3. "Qué es lo que funciona mal en mí" [What's wrong with me?]. | .67 | .75 | .72 |
| 4. Soy un inútil, no valgo para nada [I'm worthless]. | .62 | .71 | .70 |
| 5. Me siento tan impotente, tan desamparado [I feel so helpless]. | .53 | .74 | .70 |
| 6. Algo tiene que cambiar [Something has to change]. | .58 | .71 | .67 |
| 7. Mi future es un desierto [My future is bleak]. | .48 | .70 | .66 |
| 8. No consigo terminar nada de lo que empiezo [I can't finish anything]. | .38 | .59 | .55 |

**Table 2  Coefficient alpha and omega, and descriptive data across samples.**

| | Sample 1: Undergraduates (N = 302) | Sample 2: General population online (N = 846) | Overall sample (N = 1,148) |
|---|---|---|---|
| Alpha [95% CI] | .83 [.80, .85] | .90 [.89, .91] | .89 [.88, .90] |
| Omega [95% CI] | .83 [.78, .86] | .90 [.89, .91] | .89 [.88, .90] |
| Mean score (SD) | 14.46 (5.40) | 16.22 (6.80) | 15.76 (6.50) |

corrected item-total correlation ranging from .55 to .74 for the overall sample and good discrimination indices.

Table 2 presents the alpha and omega coefficients of the ATQ-8 for Samples 1 and 2. The alpha of the overall sample was .89 (95% CI [.88, .90]), whereas the omega was also .89 (95% CI [.88, .90]). Table 2 also shows the descriptive data of the ATQ-8. There were no statistically significant differences on the ATQ-8 scores between genders in Sample 1 (women: $M = 16.54$, $SD = 6.92$; men: $M = 15.16$, $SD = 6.40$). However, in Sample 2, women showed higher scores on the ATQ-8 than men (women: $M = 14.39$, $SD = 5.44$; men: $M = 14.58$, $SD = 5.35$).

### Validity evidence based on internal structure
#### Dimensionality
The one-factor model obtained an adequate fit according to the goodness-of-fit indexes: $\chi^2$ (20) = 251.202, $p < .01$; RMSEA = 0.10, 90% CI [0.089, 0.112], CFI = 0.98, NNFI = 0.97, and SRMR = 0.0483. Figure 1 presents the results obtained from the completely standardized solution of the one-factor model.

#### Measurement invariance
Table 3 displays the results of the analysis of the scalar and metric invariance. Changes in RMSEA, CFI, and NNFI were lower than .01 in all cases. Therefore, parameter invariance
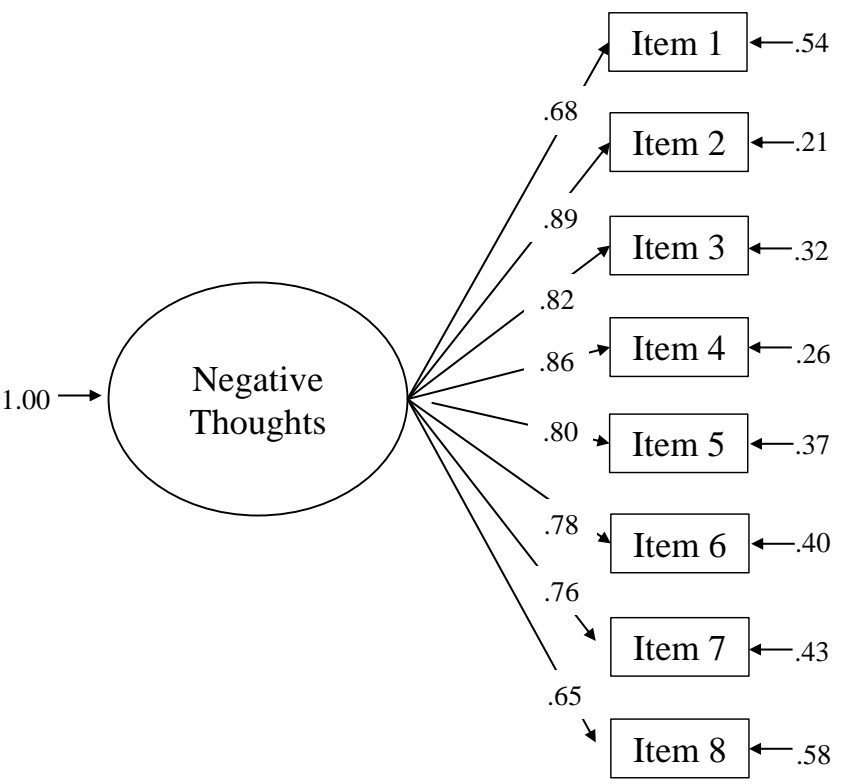

**Figure 1 Completely standardized solution of the ATQ-8 one-factor model.**

was supported at both the scalar and metric levels across samples, gender, groupage, and education level.

## Validity evidence based on relationships with other variables

Table 4 shows that the ATQ-8 presented correlations with all of the other constructs that were assessed in the expected direction: it presented positive correlations with dysfunctional schemas (DAS-R), experiential avoidance (AAQ-II), emotional symptoms (DASS-21), generalized pliance (GPQ), and cognitive fusion (CFQ); and negative correlations life satisfaction (SWLS).

## DISCUSSION

While the ATQ has already been validated in Spain, to our best knowledge, no study has analyzed the factor structure and psychometric properties of the ATQ-8 with Spanish samples. This version has two main advantages over the original ATQ. Firstly, the factor structure of the ATQ-8 is more simple and stable than the one of the original ATQ. Secondly, the ATQ-8 is better suited to survey research and provides a considerably briefer assessment than the original ATQ. Accordingly, this study aimed to explore the psychometric properties of the ATQ-8 in two Spanish samples.

**Table 3  Metric and scalar invariance across sample, gender, group age, and education level.**

| Model | RMSEA | ΔRMSEA | CFI | ΔCFI | NNFI | ΔNNFI |
|---|---|---|---|---|---|---|
| Measurement invariance across sample | | | | | | |
| MG Baseline model | .0983 | | .982 | | .975 | |
| Metric invariance | .1000 | −.0017 | .978 | - .004 | .974 | - .001 |
| Scalar invariance | .0984 | .0016 | .976 | - .002 | .975 | .001 |
| Measurement invariance across gender | | | | | | |
| MG Baseline model | .101 | | .980 | | .973 | |
| Metric invariance | .0915 | .0095 | .981 | .001 | .977 | .004 |
| Scalar invariance | .0903 | .0120 | .979 | −.002 | .978 | .001 |
| Measurement invariance across group age | | | | | | |
| MG Baseline model | .1001 | | .979 | | .971 | |
| Metric invariance | .1047 | −.0046 | .973 | −.006 | .968 | -.003 |
| Scalar invariance | .1089 | −.0042 | .966 | −.007 | .965 | -.003 |
| Measurement invariance across education level | | | | | | |
| MG Baseline model | .1013 | | .981 | | .974 | |
| Metric invariance | .1046 | −.0033 | .976 | −.005 | .972 | -.002 |
| Scalar invariance | .1033 | .0013 | .974 | −.002 | .973 | .001 |

**Table 4  Pearson correlations between the ATQ-8 scores and other relevant self-report measures.**

| Measure | S | N | r with ATQ-8 |
|---|---|---|---|
| DAS-R | 1 | 302 | .43[*] |
| DASS—Total | 1 | 302 | .60[*] |
| | 2 | 846 | .74[*] |
| DASS—Depression | 1 | 302 | .61[*] |
| | 2 | 846 | .78[*] |
| DASS—Anxiety | 1 | 302 | .47[*] |
| | 2 | 846 | .56[*] |
| DASS—Stress | 1 | 302 | .49[*] |
| | 2 | 846 | .62[*] |
| AAQ-II | 1 | 302 | .59[*] |
| | 2 | 846 | .70[*] |
| CFQ | 1 | 302 | .63[*] |
| | 2 | 846 | .65[*] |
| GPQ | 1 | 302 | .31[*] |
| | 2 | 846 | .48[*] |
| SWLS | 2 | 846 | −.63[*] |

Notes.

AAQ-II, Acceptance and Action Questionnaire-II; ATQ-8, Automatic Thoughts Questionnaire-8; CFQ, Cognitive Fusion Questionnaire; DAS-R, Dysfunctional Attitude Scale-Revised; DASS, Depression, Anxiety, and Stress Scales-21; GPQ, Generalized Pliance Questionnaire; SWLS, Satisfaction with Life Scale.

[*]$p < .001$.

The analyses indicated that the Spanish version of the ATQ-8 showed good psychometric properties in Spain. Concerning internal consistency, the ATQ-8 displayed an alpha of .89, and the items had corrected item-total correlations ranging from .38 to .74. Confirmatory

factor analyses showed that the one-factor model presented a good fit to the data as in the previous studies by *Netemeyer et al. (2002)* and *Ruiz et al. (2017b)*. Also, the ATQ-8 showed metric and scalar measurement invariance across the type of sample (undergraduates and general online population), gender, groupage (younger or equal than 35 years vs. older than 35 years), and education level (primary and secondary studies vs. university studies). These analyses indicate that the ATQ-8 scores can be compared across these variables. Additionally, the ATQ-8 demonstrated convergent validity, given the positive correlations found with emotional symptoms, dysfunctional schemas, generalized pliance, experiential avoidance and cognitive fusion, and the negative correlations with life satisfaction.

It is worth to mention some limitations of this study. Firstly, we did not collect data from a clinical sample. This is a significant limitation because the ATQ was mainly designed to assess clinical participants. Accordingly, further studies should analyze the psychometric properties of the ATQ-8 in a clinical sample and, as in *Ruiz et al. (2017b)*, to explore the measurement invariance across clinical and nonclinical samples. Secondly, as this study did not include a clinical sample, we were not able to analyze if the ATQ-8 can be used as a screening measure to detect unipolar depression. Thirdly, the psychometric properties of the ATQ-8 were analyzed in two convenience samples. Thus, the representativeness of the samples is uncertain. Accordingly, further studies should be conducted with other Spaniard samples to confirm the results of the current study. Fourthly, we did not explore the sensitivity to treatment. However, note that the study by *Ruiz et al. (2017b)* showed that the ATQ-8 was sensitive to treatment in a clinical study conducted in Colombia. Lastly, the percentage of women was significantly higher than the percentage of men in the composition of the samples. However, the finding of measurement invariance across gender reduces this limitation.

## CONCLUSIONS

The findings of the current study are consistent with previous studies by *Netemeyer et al. (2002)* and *Ruiz et al. (2017b)*. Importantly, this study adds empirical evidence of the adequate fit of the one-factor structure of the ATQ-8 and its measurement invariance across gender, age, and education level. Further studies should try to replicate these findings in other Spanish-speaking countries and analyze the measurement invariance of the ATQ-8 across different cultures and countries. In conclusion, the ATQ-8 was a reliable and valid instrument in a Spanish sample. Therefore, it seems the ATQ-8 can be used in Spain as a less time-consuming measure of negative automatic thoughts than the original ATQ.

### Funding
The authors received no funding for this work.

### Competing Interests
The authors declare there are no competing interests.

## Author Contributions

- Francisco J. Ruiz conceived and designed the experiments, analyzed the data, prepared figures and/or tables, authored or reviewed drafts of the paper, and approved the final draft.
- Miguel A. Segura-Vargas conceived and designed the experiments, prepared figures and/or tables, authored or reviewed drafts of the paper, and approved the final draft.
- Paula Odriozola-González conceived and designed the experiments, performed the experiments, authored or reviewed drafts of the paper, and approved the final draft.
- Juan C. Suárez-Falcón conceived and designed the experiments, analyzed the data, authored or reviewed drafts of the paper, and approved the final draft.

## Human Ethics

The following information was supplied relating to ethical approvals (i.e., approving body and any reference numbers):

Bioethical Institutional Committee, Fundación Universitaria Konrad Lorenz approved this study (2016-021B).

## Data Availability

The raw data are available as Supplemental File.

## Supplemental Information

Supplemental information for this article can be found online at http://dx.doi.org/10.7717/peerj.9747#supplemental-information.

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
