# Peer review of "Psychometric properties of the Automatic Thoughts Questionnaire-8 in two Spanish nonclinical samples"

_PeerJ, doi:10.7717/peerj.9747_

## Round 0.1 · original submission · Major Revisions

Thank you for submitting this psychometric study. Based on the comments of reviewers, I suggest the authors to revise their paper accordingly. No clinical sample used should be considered as a major limitation of this study. For clinical implication of ATQ-8, data from clinical samples are important. Sample representativeness is another issue, I suggest the authors to reconsider their title and tone down some terms, be specific and accurate.

Reviewer 1 ·

Basic reporting

The paper is well-written.
Literature review is sufficient.

Experimental design

A psychometric study with two samples of Spainiard people.
Focusing on criterion validity and internal consistency and construction validity. In addition, the authors also tested measurement invariance across samples and genders.

Validity of the findings

I have the below comments for the authors.
1. Sample representativeness. Two samples from undergraduate students and online general population are not representative of Spaniard people. Please consider the potential impact of sample selection on the psychometric property of ATQ-8. Without a representative sample, I think the authors can not make conclusions in their way.
2. The questionnaire contained over 80 items, it seems too long for participants. Validity of answers in the questionnaire must be considered.
3. For online survey, I do not think the authors can obtained written informed consent.
4. The basic demographic characteristics of the study samples must be reported.
5. Measurement invariance across age and education level is also important.

Reviewer 2 ·

Basic reporting

no comment

Experimental design

In terms of study design, should researchers consider conducting exploratory factor analysis of the questionnaire first, and then comparing other competitive models that may exist in confirmatory factor analysis?

Validity of the findings

The factor structure and psychometric properties of the ATQ-8 have been analyzed in two countries. The manuscript is just a simple replication of the existing research, I can't see its contribution to the literature.

Additional comments

This study aims to analyze the validity of the ATQ-8 in Spanish samples. But the factor structure and psychometric properties of the ATQ-8 have been analyzed in two countries. As pointed out at the end of the manuscript, since this study did not include clinical samples, we were unable to analyze whether ATQ-8 could be used as a screening measure to detect unipolar depression.

---

## Round 0.2 · Minor Revisions

There are still some minor issues that need to be addressed.

Reviewer 1 ·

Basic reporting

See my comments as below.

Experimental design

See my comments as below.

Validity of the findings

See my comments as below.

Additional comments

1. Abstract. Method should specify other scales used in this study and the purpose of these scales. Main statistical methods should also be reported.
2. No a separate paragraph to describe the conclusion of this study. For conclusions, Spanish samples is broad, which should be detailed here.

---

## Round 0.3 · accepted · Accept

I think the paper can be accepted now.